# A Study on the Impact of Unconventional (and Conventional) Drilling on Housing Prices in Central Oklahoma

**Kangil Lee** [1,*] and **Brian Whitacre** [2]

1   Department of Global Business, Changwon National University, Changwon 51140, Korea
2   Department of Agricultural Economics, Oklahoma State University, Stillwater, OK 74074, USA;
    brian.whitacre@okstate.edu
*   Correspondence: leekangil@changwon.ac.kr

**Abstract:** Shale energy development activity may benefit some aspects of a regional economy (such as increased jobs or tax revenue); however, there may also be negative impacts to the local environment, such as noise and underground water contamination. We study the impact of unconventional drilling activity on housing price in an area of the country with a long history of crude oil production. A prospective home buyer may want to avoid a place near sites that have been drilled using unconventional drill technologies such as horizontal fracturing. Adopting a hedonic price model, we estimate the impact of distance to and density of unconventional drilling on housing prices in two central counties in Oklahoma during the period 2001–2016. We also apply a semiparametric approach to deal with the possibility that the relationship between an environmental pollutant source and housing price is nonlinear. The empirical results are consistent in terms of physical housing characteristics and locational aspects in all cases, with drilling activity having only a minimal effect in benchmark models. Further, the semiparametric estimation results support the findings that drilling activity has only limited impacts on local housing prices.

**Keywords:** shale gas; hedonic analysis; housing values; environmental costs





## 1. Introduction

Several states across the U.S., including Oklahoma, have experienced a dramatic increase in unconventional shale gas drilling since the mid-2000s. From a regional economics perspective, this creates additional employment opportunities and likely generates positive externalities across specific industries (for example, retail and construction). A significant amount of research has documented the regional economic impacts associated with the rise in drilling, including those for employment and income [1–7]. Alternatively, several studies have examined negative externalities associated with the drilling increase, such as exacerbated educational attainment or declines in well-being in regions with high levels of drilling activity [8,9].

This issue is also of importance to the local housing market. Negative externalities generally lower the value of a property [10] and nearby unconventional drilling can be troubling to potential buyers. There is significant concern that hydraulic fracturing contaminates underground water [11–13]. Other negative impacts that may arise due to shale exploration include increased noise and traffic near the drilling site [14]. Thus, prospective home buyers may choose to avoid options near sites that have been drilled using unconventional technologies. If this is the case, this preference should reveal itself in the house price. Several recent studies have found this to be the case. To date, at least, four papers [11,12,14,15] have examined this negative externality from energy development using a hedonic analysis.

Most of this previous literature focuses on the shale gas drilling impact on house prices in Pennsylvania, and one study examined the same effect in Texas. Oklahoma has a large amount of unconventional drilling activity (like Pennsylvania); alternatively, the

state has a strong history of conventional oil production (like Texas). However, to our best knowledge, there are no studies on this topic for Oklahoma, despite warnings that the consideration of differences across study areas is important for policy makers [12]. Considering this, the aim of this study is to fill in this research gap and estimate the impact of unconventional shale drilling on housing prices in Oklahoma.

*Literature Review*

How environmental quality influences housing price is a popular topic in hedonic analysis. In particular, newly introduced energy development activities—wind turbine installation and unconventional shale drilling—have recently attracted researchers' attention. On the whole, energy development activities generally have a negative impact on house prices. Our discussion starts with two studies [16,17] that examine the negative effect of wind farms on nearby house prices. The paper [16] noted people's concerns with the negative visual impacts from a nearby wind farm. His study used housing transaction data from England and Wales during 2001–2012; however, unlike traditional hedonic analysis, the inclusion of housing characteristics was optional and, instead, the focus was on the visual impact from wind farms to house prices. The study [17] explored the same topic, using house transaction data in Dutch for the years 1985–2011. The results of their study support those from [16], with both finding that wind farm visibility negatively influenced nearby housing prices.

New shale drilling activities (particularly horizontal hydraulic fracturing) incur concerns about noise and water contamination, and economic theory suggests that such environmental disutility would lower nearby housing prices. Several studies examined these negative externalities from shale gas development by applying hedonic analysis to the local housing market [11,12,14,15]. Among those, three papers [11,12,14] focus on a county (or counties) in Pennsylvania, but their results are varied. The paper [14] is the first published work examining the shale drilling boom's effect on house prices. Applying housing transaction data from Washington County in Pennsylvania (part of the Marcellus Shale), this study is performed using standard hedonic analysis. They consider housing characteristics (such as area, number of rooms, etc.) and use geographical data on the permitting and actual drilling of shale wells and land use of the surrounding area. They found that property values are negatively impacted by nearby shale gas exploration activity, with larger effects for households that use private well water. One limitation of their analysis is that their housing transaction dataset runs from 2008 to 2010. It is hard to argue that their analysis fully encompasses the pre and post shale boom period, given the dramatic shale drilling increases starting in the mid-2000s. The study [11] considers a more comprehensive geographical area (36 counties in Pennsylvania) and time period (1995–2012) and again shows that the results are heavily dependent on the source of water for the home. They found that the impact of a well is large and negative for nearby groundwater-dependent homes, while a small and positive impact is found for piped-water-dependent homes. This implies that people are concerned about water contamination caused by shale drilling, and this concern is captured in housing prices located near shale drilling sites. The paper [12] studied this same issue, considering two counties in Pennsylvania with varying data availability (one county spanned 2006–2012, the other spanned 2004–2013). They use regression, matching, and semiparametric techniques to show that a robustly significant negative effect does not exist for unconventional gas wells on property values—thus differing from the findings of [11,14]. Finally, the study [15] empirically shows that unconventional drilling has a negative influence on housing prices for 2005–2011 in Tarrant County, Texas. Their estimates show that proximity to an unconventional well reduces housing value by 1.3–3.5%. They also estimate impacts associated with conventional well activity but find no statistically significant effects. This difference between conventional and unconventional impacts on the local housing market is an important part of their analysis.

In sum, relatively few papers provide empirical evidence that environmental concerns caused by unconventional shale drilling lower house prices. Three studies focused on

Pennsylvania in the Marcellus shale play region, and one paper considered a single county in Texas. To date, there is no study on this issue for Oklahoma, even though the state of Oklahoma is one of the most heavily drilled regions for unconventional wells. This paper attempts to fill in this research gap. Referencing previous literature and the economic theory of negative externalities, we expect that house prices may be lowered when they are located near well sites drilled via hydraulic fracturing technology. However, we need to consider the attitudes and characteristics of the local population as well. Compared to Pennsylvania, Oklahoma has a long history of fossil fuel mining, with residents accustomed to seeing traditional wells and ongoing well activity in everyday life. Thus, it is possible that people in Oklahoma are less apprehensive to new energy development exploration. Thus, the sign of the effect is theoretically ambiguous.

## 2. Study Region: Canadian County and Payne County in Oklahoma

Oklahoma is one of the leading states in the nation for unconventional drilling activity. During 2001–2016, a total of 4644 shale wells were drilled across the state (Oklahoma Geological Survey [18]). Twelve counties (Canadian, Pittsburg, Hughes, Coal, Grady, Carter, Blaine, Wagoner, Payne, Stephens, Marshall, and Logan) had more than 100 shale well sites drilled during this time. Six counties (Atoka, Garvin, Johnston, Noble, Garfield, and Dewey) had more than 50 wells. Twenty-five counties did not have any. In this study, we consider two of these 'high activity' counties—Canadian and Payne. The population of Canadian county was 116,332 and the population of Payne County was 77,448 in 2010. Both counties are located in central Oklahoma (Figure 1). Canadian County is one of seven counties in the Oklahoma City metropolitan area. A total of 673 shale wells were drilled during the period of analysis, giving the county the highest shale well density in Oklahoma (Table 1). Canadian County is composed of part of 1 major city (Oklahoma City, OK, USA) and 10 other communities, including medium-sized cities such as Mustang, El Reno, and Yukon. Most of these are located in the eastern portion of the county. Our housing transaction data (from the county assessors' office) show that purchases took place close to these 10 communities. Alternatively, most of the drilling activity took place in a more remote western portion of the county (Figure 2).

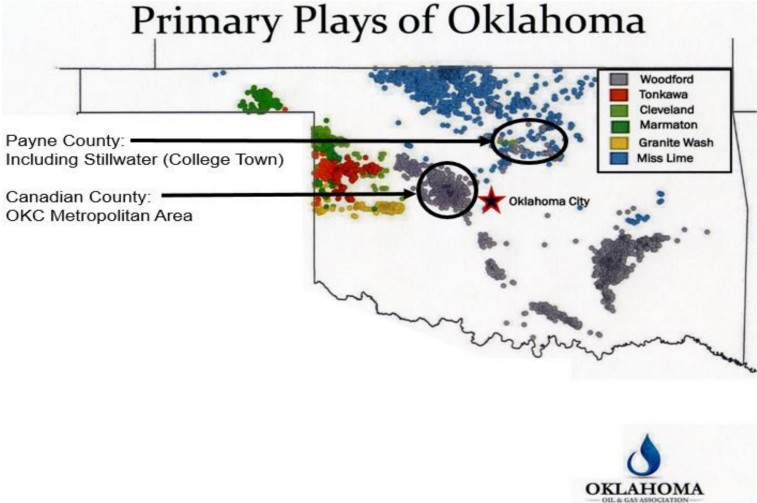

**Figure 1.** Overview of shale region in Oklahoma and location of the study region.

**Table 1.** Number of unconventional wells and housing transactions.

| Year | Canadian County | | Payne County | |
|---|---|---|---|---|
| | **Wells** | **House Transactions** | **Wells** | **House Transactions** |
| 2001 | 0 | 1 | 0 | 222 |
| 2002 | 0 | 3 | 0 | 285 |
| 2003 | 0 | 0 | 0 | 293 |
| 2004 | 0 | 4 | 0 | 424 |
| 2005 | 0 | 2 | 0 | 451 |
| 2006 | 0 | 5 | 0 | 480 |
| 2007 | 1 | 5 | 0 | 493 |
| 2008 | 20 | 1747 | 0 | 509 |
| 2009 | 37 | 1830 | 0 | 529 |
| 2010 | 59 | 1579 | 2 | 508 |
| 2011 | 77 | 1559 | 1 | 510 |
| 2012 | 132 | 1813 | 14 | 728 |
| 2013 | 135 | 2104 | 22 | 888 |
| 2014 | 85 | 2324 | 77 | 1012 |
| 2015 | 82 | 2344 | 40 | 1093 |
| 2016 | 45 | 2437 | 17 | 1305 |
| Total | 673 | 17,757 | 173 | 9730 |

Note: Numbers of house transactions are based on after-data trimming of assessor's office raw data. Sources: Drilled Wells (the Oklahoma Geological Survey); House Transactions (the assessors' office in each county).

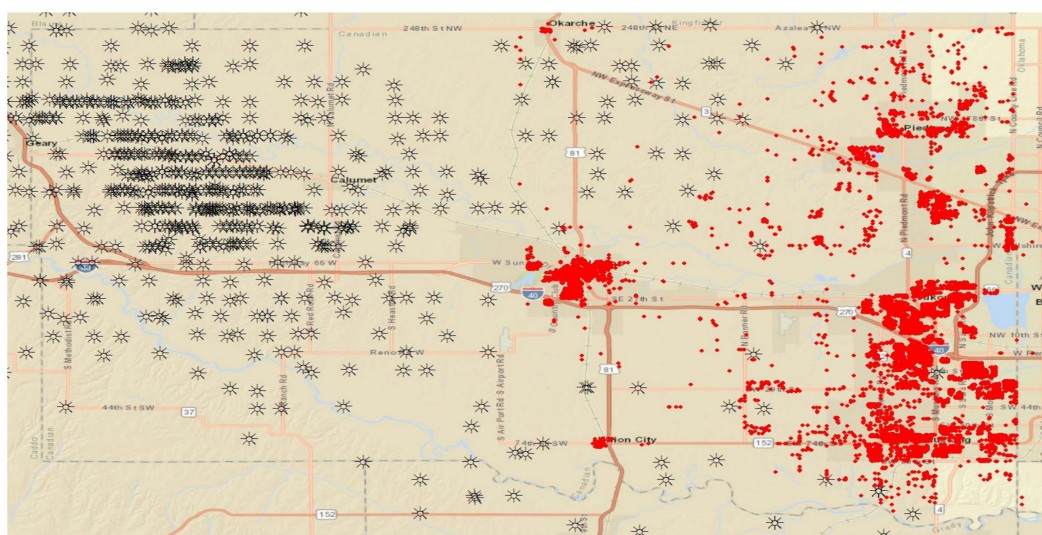

**Figure 2.** Locations for house transaction and unconventional drilling well sites in Canadian County. Note: ✷ indicates shale drilling site and ● indicates the location of traded house. Source: Drilled Well (the Oklahoma Geological Survey); House Transactions (the Canadian County assessors' office).

Payne County had 173 shale wells from 2001–2016, making it the 9th highest shale well-dense county in Oklahoma. Stillwater and Cushing are the biggest cities in the county and these two cities generate about 81% of all total housing transactions. Counter to the pattern seen in Canadian County, however, the well sites and residential area have a much higher degree of overlap (Figure 3). This can be verified with the data in Table 2. The mean value of the number of drillings within a specific radius is larger for a house in Payne County than for one in Canadian County. Similarly, the mean value of the distance between a house and the nearest shale well in Payne County is smaller than its counterpart in Canadian County. Thus, these two counties offer significantly different scenarios for assessing the relationship between drilling site and housing transaction in Oklahoma.

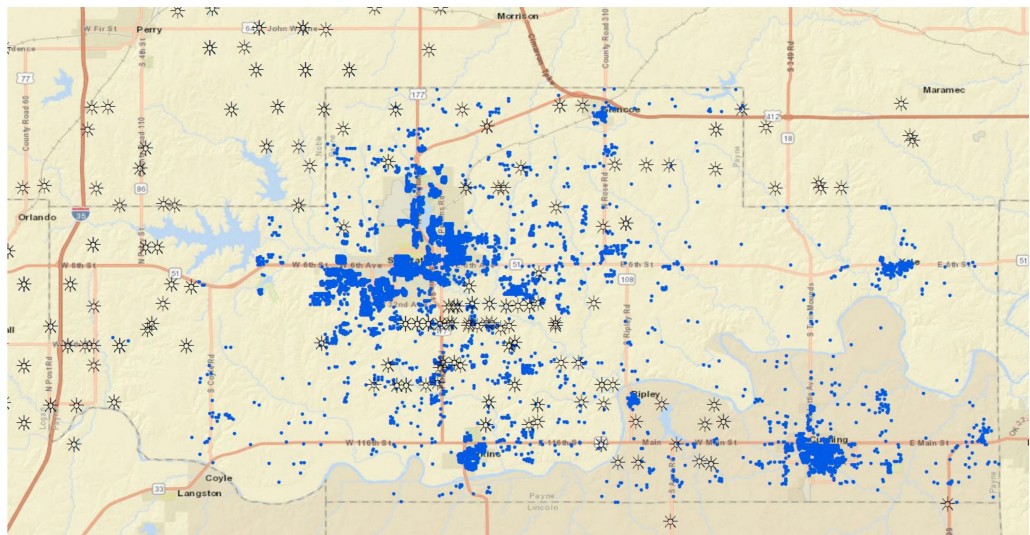

**Figure 3.** Locations for house transaction and unconventional drilling well sites in Payne County. Note: ☼ indicates shale drilling site and ● indicates the location of traded house. Source: Drilled Well (the Oklahoma Geological Survey); House Transactions (the Payne County assessors' office).

**Table 2.** Summary statistics.

| Variable | Canadian County | | | | | Payne County | | | | |
|---|---|---|---|---|---|---|---|---|---|---|
| | Obs | Mean | Std. Dev. | Min | Max | Obs | Mean | Std. Dev. | Min | Max |
| Public Water | 17,757 | 0.93 | 0.26 | 0 | 1 | 9730 | 0.81 | 0.39 | 0 | 1 |
| Distance to the Nearest Road | 17,757 | 417.23 | 409.14 | 0.00 | 4152.35 | 9730 | 383.73 | 440.22 | 3.79 | 3157.57 |
| Distance to the Nearest Highway | 17,757 | 1155.47 | 1233.68 | 0.03 | 10,652.10 | 9730 | 1136.68 | 1132.32 | 14.12 | 7172.08 |
| Distance to the Nearest Unconventional Drill Site | 17,748 | 12.63 | 6.24 | 0.26 | 47.87 | 7940 | 8.30 | 8.60 | 0.09 | 40.87 |
| Count in Ring I (Unconventional Drilling) | 17,757 | 0.00 | 0.02 | 0 | 1 | 9730 | 0.04 | 0.32 | 0 | 8 |
| Count in Ring II (Unconventional Drilling) | 17,757 | 0.00 | 0.02 | 0 | 1 | 9730 | 0.04 | 0.33 | 0 | 6 |
| Count in Ring III (Unconventional) | 17,757 | 0.00 | 0.02 | 0 | 1 | 9730 | 0.09 | 0.57 | 0 | 10 |
| Distance to the Nearest Conventional Drill Site | 17,757 | 1.85 | 1.20 | 0.01 | 8.25 | 9730 | 2.21 | 1.12 | 0.03 | 6.98 |
| Count in Ring I (Conventional Drilling) | 17,757 | 0.16 | 0.44 | 0 | 5 | 9730 | 0.07 | 0.41 | 0 | 10 |
| Count in Ring II (Conventional Drilling) | 17,757 | 0.15 | 0.42 | 0 | 5 | 9730 | 0.10 | 0.37 | 0 | 6 |
| Count in Ring III (Conventional Drilling) | 17,757 | 0.21 | 0.52 | 0 | 6 | 9730 | 0.14 | 0.44 | 0 | 6 |
| Sale Price | 17,757 | 159,960.7 | 169,233.5 | 500 | 4,700,000 | 9730 | 125,332.3 | 126,248.7 | 25.00 | 6,100,000 |
| Age of Bldg | 17,757 | 25.88 | 29.44 | 0 | 2010 | 8628 | 34.78 | 28.31 | 0 | 120 |
| # of Bath | 17,603 | 1.95 | 0.55 | 0 | 6 | 9727 | 1.71 | 0.93 | 0 | 8 |
| # of Beds | 17,562 | 3.10 | 0.75 | 0 | 38 | 8603 | 3.04 | 0.94 | 0 | 12 |
| Area | 17,757 | 1754.66 | 664.64 | 72 | 8585 | 8618 | 2162.29 | 1036.99 | 0 | 11,011.16 |

Note: Unit for distance variables are mile and for sale price is current USD. Sources: Drilled Wells (the Oklahoma Geological Survey and the Oklahoma Corporation Commission); House Transactions (the assessors' office in each county); Road Information (the Oklahoma Department of Transportation); Public water accessibility (the Oklahoma water resource board).

## 3. Methodology

### 3.1. Hedonic Price Model

To examine how unconventional shale drilling impacts the housing price in our two counties, we adopt a hedonic price model. Hedonic analysis is the most commonly used empirical methodology to study housing price evaluation. The spirit of hedonic analysis is that the value of an item can be estimated using its characteristics. Further, an implicit

price of each attribute corresponds to the market equilibrium price for that attribute. This can be verified using a classical microeconomic utility maximization framework.

Consider the following hedonic price function: $P_i = f(H_i, Q_i)$, where $P_i$ refers to the price of the $i^{\text{th}}$ house, $H_i$ refers to the characteristics of the $i^{\text{th}}$ house (such as number of rooms, number of bathrooms, age, and so on), and $Q_i$ implies a quality of neighborhood environment for the $i^{\text{th}}$ house (in this study, environmental quality represents the relationship between a house and an unconventional or conventional drilling site). Then $\frac{\partial P_i}{\partial Q_i}$ can be regarded as the marginal implicit price of the house with respect to environmental quality.

### 3.2. Empirical Estimation Strategy: Benchmark Model

Regarding the estimation of the hedonic price function, there is no consensus on the specific functional form to use. Further, the study [19] found that the true hedonic functional form is generally unknown. Despite limited guidance on this issue, several different transformation equations are commonly used, including the semi-log, double log, and Box-Cox. In our case, the semi-logarithmic equation is preferred over double log form given our inclusion of a significant number of dummy variables. Thus, we start with the semi-logarithmic equation as our benchmark model. Then, we estimate a hedonic price using the specification:

$$\ln(P_{it}) = \sum_j \alpha_j X_{itj} + \beta Drilling_{it} + \gamma PW_{it} + \delta Dist_{it} + \zeta_t + \eta_c + \varepsilon_{it} \tag{1}$$

where the variable $X_{it}$ includes control variables in terms of housing characteristics such as the number of bedrooms, number of bathrooms, area square feet, and age of building; $Drilling_{it}$ denotes variables associated with drilling activity near house $i$; $PW_{it}$ is a dummy that takes a value of one if the house has public water supply available to it; $Dist_{it}$ includes controls for the distance between a house and major road or highway; $\zeta_t$ are time dummies—specifically, year fixed effects; and $\eta_c$ are community fixed effects. A community is defined as a city or town in the two counties. For Canadian County, there are seven community districts: El Reno; Mustang; Okarche, Oklahoma City; Piedmont; Union City; and Yukon. For Payne County, there are sixteen community districts: Coyle, Cushing (rural), Cushing (town), Drumright, Glencoe (rural), Glencoe (town), Morrison, Oak Grove, Perkins (rural), Perkins (town), Ripley (rural), Ripley (town), Stillwater (rural), Stillwater (town), Yale (rural), and Yale (town). These communities are defined by each county assessor.

To examine the impact of drilling on housing price, previous studies [11,12,14,15] use a count variable, called a ring, to represent the number of wells within a specific distance (i.e., a density measure). Alternatively, two others [11,12] also consider how far the nearest well is located from a house (i.e., a distance measure). In this study, we adopt both types of variables, distance and density, to measure drilling effects. We can then define two distinct versions of our benchmark model:

$$\ln(P_{it}) = \sum_j \alpha_j X_{itj} + \rho D_{it} + \gamma PW_{it} + \delta Dist_{it} + \zeta_t + \eta_c + \varepsilon_{it} \tag{2}$$

$$\ln(P_{it}) = \sum_j \alpha_j X_{itj} + \sigma CNT_{it} + \gamma PW_{it} + \delta Dist_{it} + \zeta_t + \eta_c + \varepsilon_{it} \tag{3}$$

where $D_{it}$ refers to the distance between house $i$ and its nearest shale drilling well location and $CNT_{it}$ refers to the number of drillings within a specific radius from house $i$. We consider three radius criteria (0–3500, 3501–5000, and 5001–6500 feet). Previous studies introduced similar distance standards: the study [11] uses 1 and 1.5 km, while the paper [14] considered 0.75 and 2 miles. The range of [12] is 1-4 miles. The other paper [15] uses 0–3500, 3501–5000, and 5001–6500 feet.

### 3.3. Additional Estimation: Semiparametric Estimation

To verify the robustness of the benchmark model results, we incorporate a semiparametric estimation approach. Although the physical housing characteristics are usually well explained by a linear relationship, some attributes may not be identified by linear estimation. For example, the paper [20] argues that nonparametric estimations may better explain a hedonic price model due to the existence of nonlinearities between house prices and their associated attributes. The study [21] demonstrates that semiparametric modelling shows good robustness, although these techniques do have some disadvantages, such as lesser precision and the need for a high number of observations (known as the curse of dimensionality). The study [22] applied a semiparametric approach to estimate a hedonic price function. They argue that the semiparametric approach shows better empirical results than the parametric model, with more accurate mean predictions. Returning the effects of unconventional drilling on housing prices, the paper [12] applied this semiparametric specification and showed that a nonlinear relationship exists between housing price and unconventional drilling sites. Applying this empirical strategy, traditional physical housing characteristics such as the number of bedrooms or bathrooms are specified in the linear part in the model, while the variables for unconventional well activity are modelled in the nonlinear portion. This allows us to avoid potentially misspecifying the nonparametric portion of the regression equation.

To model a partially linear specification as our robustness test strategy, consider a following generalized partial linear equation,

$$y = X\Gamma + f(z) + \varepsilon \tag{4}$$

where $y$ denotes house price, $X$ is a control variable vector for physical housing attributes (e.g., number of rooms, age of building, etc.), and $z$ is an impact from a drilling site. In this partially linear model, the unknown function $f(z)$ is determined by the data, and the rest of the controls are specified linearly. To estimate this, we adopt Robinson's double residual model [21]. The derivation of the double residual model estimator starts with taking a conditional expectation (only for $Z$) in Equation (4):

$$E(y|Z) = E(X|Z)\Gamma + f(z) + E(\varepsilon|Z) \tag{5}$$

Combining Equations (4) and (5) leads to

$$y - E(y|Z) = (X - E(X|Z))\Gamma + \varepsilon. \tag{6}$$

An advantage of this approach is that the unknown function $f(z)$ can be removed as Equation (6) demonstrates. It can then be restated as,

$$\widetilde{y} = \widetilde{x}\Gamma + \widetilde{\varepsilon}, \text{ where } \widetilde{a} = [a - E(a|Z)] \tag{7}$$

Equation (7) is exactly a linear equation form. That is, $\Gamma$ is a consistent estimator, and we can estimate it by using a traditional OLS estimator Thus,

$$\hat{\Gamma} = \left(\widetilde{\varepsilon_2}'\widetilde{\varepsilon_2}\right)^{-1}\widetilde{\varepsilon_2}'\widetilde{\varepsilon_1}. \tag{8}$$

Then, by the nonparametric regression estimation for each error term ($\varepsilon_1$ and $\varepsilon_2$), and replacing those estimates into Equation (8), $\hat{\Gamma}$ can be estimated (note that $\varepsilon_1$ implies $(y - E(y|Z))$ and $\varepsilon_2$ implies $(X - E(X|Z))$). Finally, we can obtain unknown function $f(z)$ by nonparametric regression $(y - X\hat{\Gamma})$ on $Z$.

## 4. Data

### 4.1. House Transaction Data

Our house transaction data were provided by the county assessor's office from both Payne and Canadian counties. Each dataset is composed of the characteristics of each

house sold, sales price, and date of sale, and precise geographic location (address or latitude/longitude coordinates). The housing characteristics include attributes such as the number of bedrooms and bathrooms, the year in which the house was built, living area in square feet, indicators for town of residence, and indicators for basement, garage, fireplace, and central air. The time period of the dataset for both counties covers the years from 2001 to 2016. Note that the number of housing transactions in the data received is minimal for Canadian County prior to 2008. Since the cumulative number of unconventional wells was only 1 as of 2007, this should not affect the analysis. This time span encompasses the pre and post shale gas drilling period (the peak drilling years were 2013 and 2014 from Table 1). The raw data from the county assessor's office contained some observations with missing data or inappropriate location information. After removing these observations, our housing transaction sample for Canadian and Payne counties have 17,757 and 9730 observations, respectively. Summary statistics are represented in Table 2. Average housing prices are larger in Canadian County (USD 160,000 vs. 125,000), and houses are typically newer (average age 26 years vs. 35 in Payne). The average house in each county has three bedrooms and less than two bathrooms.

### 4.2. Drilling and the Other Geographical Data

Historical unconventional drilling data were obtained from the Oklahoma Geological Survey [18]. This dataset includes precise geographical information (latitude and longitude coordinates) and date information for the completion of each shale drilling site. Using this information, a distance between a house and each well is calculated. The distance information is included in the regressions only when the shale gas well was drilled prior to the housing transaction date. Under our modeling approach, this variable reflects how housing price is influenced by the distance to the nearest drilling site. A positive and significant parameter would suggest that housing price increases with the distance to the closest well—which might be expected if potential buyers have concerns about how such wells impact their quality of life. Additionally, the number of drilling sites within a specific radius from a house are counted using this distance information. Previous studies applied different, but relatively similar, distance standards. Following [15], we applied three different distances cases (0–3500, 3501–5000, and 5001–6500 feet). (Unlike the other three previous papers on Pennsylvania locations, the paper [15] studied the Dallas Fort Worth area, which has a long oil mining history, similar to Oklahoma). These variables can explain how housing price might be influenced by the density of drilling near a house's location.

We also consider conventional wells in our analysis to assess if differences exist between the impacts for conventional and unconventional wells. Historical conventional drilling data were gathered from the Oklahoma Corporation Commission (OCC) [23]. The dataset from OCC contains all drilling logs from wells drilled in the state since 1900. We eliminate unconventional drilling (Directional Hole and Horizontal Hole) observations from the well completion master file to construct a conventional well dataset. Note that, in both counties, the average distance from a house to a conventional well is significantly lower than the distance to an unconventional well (Table 2).

The other spatial data components included in our baseline regressions are information on the distance between a house and a major highway and local road and the accessibility of public water supply. Using a shapefile from the Oklahoma Department of Transportation [24], a highway distance is calculated for each house in ArcGIS software. A map of public water accessibility was created using a shapefile from the Oklahoma Water Resource Board [25]. The corresponding variable takes on a value of one if the housing location overlaps with public water provision within the state. Roughly 93% of transacted houses in Canadian County are located inside the public water supply region (Table 2). That is, about 93% of houses in Canadian County have public water accessibility. In Payne County, 81% of houses are connected to the public water supply. This variable is important because of environmental concerns about drilling sites and possible water contamination. If a property is located on a site not served by public water, there may be more concern



about the impact of nearby drilling (since the property likely maintains their own well). In particular, two studies [11,14] both found that the negative effects of shale drilling were highest for houses served by private water sources.

## 5. Results and Discussion

As described in the methodology section, our estimation model includes controls for physical characteristics of the house, information related to location such as distance to the nearest highway or major city, and the availability of public water. In general, the resulting estimates accord closely with theoretical expectations. For instance, more bedrooms (or bathrooms) result in higher sale prices, while older houses result in lower prices. Further, both Canadian and Payne counties show very similar estimation results with each other regarding the returns to specific household characteristics. Overall, the estimation results generally suggest that unconventional well impacts are not significant. We will document the case that considers community-level fixed effects for each of the two counties.

### 5.1. Baseline Regression: Housing Characteristics and Locational Components

The baseline regressions are estimated via the models in Equations (2) and (3). The housing transaction data include some suspicious entries, such as extremely low purchase prices (e.g., minimum values for housing prices are USD 500 in Canadian County and USD 25 in Payne County). We suspect that some of these transactions were not appropriately reported or, at a minimum, are not reflective of a house's true value (perhaps a house was sold to a family member at a significantly reduced price). Considering this, we estimate our baseline regression models excluding observations with housing prices less than USD 5000. In addition, we remove observations with no bedrooms since these entries are also regarded as suspicious. Therefore, the number of observations reported in the model results will be lower than those in Table 2. Baseline regression results are represented in Table 3 (for unconventional drilling wells) and Table 4 (for conventional drilling wells).

In general, the parameter estimates on the physical characteristics of a house are consistent with results from most hedonic studies dealing with housing prices. Increasing the number of bedrooms and bathrooms raises the house price, other things being equal. More specifically, one extra bedroom leads to an increase of 2.8–3.5% in housing prices for Canadian County and 3.3–4.1% in Payne County. Similarly, one extra bathroom increases the housing price 15.9–16.6% in Canadian County and 13.2–15.7% for Payne. On the other hand, the sale price declines by around 0.6% in Canadian County and 0.4% in Payne County with each additional year of age, so that older houses lead to lower prices (ceteris paribus). If a house is located in an area with publicly provided water, the price of the house is roughly 3% higher than others in Canadian County. In Canadian County, about 93% of houses are linked to a public water source. Alternatively, the estimation results demonstrate that there is no impact of public water supply on the housing price in Payne County, where the public water supply rate is 81%. One possibility is that, since rural wells are relatively more common in Payne County, no premium exists for public water. It could also be that the rural water districts serving the two counties have different reputations and associated expectations. Still, this difference between Payne and Canadian County is notable, and requires additional insight.

Estimation results for control variables regarding geographical distances to large cities or highways are also interesting. In general, the results for this category of independent variables are very similar between the two counties, with one exception. This exception is the distance to the center of the largest city in the county: Stillwater (Payne County) or Oklahoma City (Canadian County). The Payne County results suggest that having a house located further away from Stillwater results in a higher housing price, while the Canadian County model finds that housing prices are not affected by their distance to Oklahoma City. This may reflect the fact that Stillwater is a college town, with a heavy percentage of houses located in the city limits dedicated to student living. Houses located just outside of the town (in suburban Stillwater) would be larger and are typically dedicated to family-oriented

residences. This may lead to an increase in housing price as houses are located farther away from the center of the town. A positive sign for the non-squared term on distance to highway or the nearest road implies that housing price increases with distance, ceteris paribus. However, it is possible that this relationship is nonlinear. For example, highways very close to a house may result in noise and traffic congestion that reduce welfare, and so some people prefer not to live very close to a highway. Alternatively, if highways are located too far from a house, travel to other towns and regions becomes too inconvenient. In addition, most major highways or interstates go through big cities rather than small towns. Considering these issues, the positive sign of the linear term and the negative sign for the quadratic term for distance to a highway are reasonable results. Note, however, that their value is very small (less than 0.1 % per mile in all cases).

**Table 3.** Benchmark model results for unconventional drilling well.

| Ln (Sale Price) | Canadian County | | | | Payne County | | | |
|---|---|---|---|---|---|---|---|---|
| | **(1)** | **(2)** | **(3)** | **(4)** | **(5)** | **(6)** | **(7)** | **(8)** |
| Distance to nearest drilling site | 0.001 (0.002) | - - | - - | - - | −0.003 (0.003) | | | - - |
| Ring Boundary I (0–3500 ft) | - - | −0.591 (0.551) | - - | - - | - - | 0.008 (0.032) | | - - |
| Ring Boundary II (3501–5000 ft) | - - | - - | 0.203 (0.163) | - - | - - | | 0.021 (0.034) | - - |
| Ring Boundary III (5001–6500 ft) | - - | - - | - - | **−0.269 *** (0.150) | - - | | | −0.009 (0.017) |
| Bedrooms | **0.034 *** (0.012) | **0.035 *** (0.012) | **0.034 *** (0.012) | **0.028 ** (0.012) | **0.041 ** (0.016) | **0.033 ** (0.015) | **0.033 ** (0.015) | **0.033 ** (0.015) |
| Bathrooms | **0.166 *** (0.040) | **0.166 *** (0.040) | **0.166 *** (0.040) | **0.159 *** (0.038) | **0.157 *** (0.020) | **0.132 *** (0.020) | **0.132 *** (0.020) | **0.132 *** (0.020) |
| Age of Building | **−0.006 ** (0.003) | **−0.006 ** (0.003) | **−0.006 ** (0.003) | **−0.006 ** (0.003) | **−0.004 *** (0.000) | **−0.004 *** (0.000) | **−0.003 *** (0.000) | **−0.004 *** (0.000) |
| Area | **0.000 *** (0.000) | **0.000 *** (0.000) | **0.000 *** (0.000) | **0.000 *** (0.000) | **0.000 *** (0.000) | **0.000 *** (0.000) | **0.000 *** (0.000) | **0.000 *** (0.000) |
| Public Water Supply | **0.032 ** (0.014) | **0.032 ** (0.013) | **0.032 ** (0.013) | **0.024 ** (0.012) | 0.026 (0.053) | 0.057 (0.048) | 0.058 (0.047) | 0.053 (0.047) |
| Distance to Biggest City (OKC or Stillwater) | 0.007 (0.014) | 0.003 (0.014) | 0.006 (0.014) | 0.006 (0.013) | **0.044 *** (0.011) | **0.051 *** (0.010) | **0.051 *** (0.010) | **0.051 *** (0.010) |
| Distance to biggest city_sq | 0.000 (0.000) | 0.000 (0.000) | 0.000 (0.000) | 0.000 (0.000) | 0.001 (0.001) | 0.000 (0.001) | 0.000 (0.001) | 0.000 (0.001) |
| D_Nearest Highway | **0.000 *** (0.000) | **0.000 *** (0.000) | **0.000 *** (0.000) | **0.000 *** (0.000) | **0.000 *** (0.000) | **0.000 *** (0.000) | **0.000 *** (0.000) | **0.000 *** (0.000) |
| D_Highway_sq | **−0.000 *** (0.000) | **−0.000 *** (0.000) | **−0.000 *** (0.000) | **−0.000 *** (0.000) | **−0.000 *** (0.000) | **−0.000 *** (0.000) | **−0.000 *** (0.000) | **−0.000 *** (0.000) |
| D_nearest_road | **0.000 *** (0.000) | **0.000 *** (0.000) | **0.000 *** (0.000) | 0.000 (0.000) | −0.000 (0.000) | −0.000 (0.000) | −0.000 (0.000) | −0.000 (0.000) |
| D_road_sq | −0.000 (0.000) | −0.000 (0.000) | −0.000 (0.000) | −0.000 (0.000) | −0.000 (0.000) | −0.000 (0.000) | −0.000 (0.000) | −0.000 (0.000) |
| Year FE | Y | Y | Y | Y | Y | Y | Y | Y |
| Community FE | Y | Y | Y | Y | Y | Y | Y | Y |
| R-sq | 0.596 | 0.596 | 0.596 | 0.602 | 0.442 | 0.457 | 0.457 | 0.457 |
| adj. R-sq | 0.595 | 0.595 | 0.595 | 0.601 | 0.439 | 0.454 | 0.454 | 0.454 |
| N | 16,848 | 16,850 | 16,850 | 16,785 | 6730 | 8227 | 8227 | 8227 |

Robust Standard errors in parentheses: * *p* < 0.1, ** *p* < 0.05, *** *p* < 0.01. Note: Observations with housing prices less than USD 5000 were removed. Dark background color in the table represents the coefficients of interest (i.e., those associated with distance from housing to drilling site). Bold text represents statistically significant results.

**Table 4.** Benchmark model results for conventional drilling well.

| Ln (Sale Price) | Canadian County | | | | Payne County | | | |
|---|---|---|---|---|---|---|---|---|
| | (1) | (2) | (3) | (4) | (5) | (6) | (7) | (8) |
| Distance to nearest drilling site | −0.006 ** | - | - | - | 0.039 *** | - | - | - |
| | (0.003) | - | - | - | (0.008) | - | - | - |
| Ring Boundary I (0–3500 ft) | - | 0.008 | - | - | - | 0.004 | - | - |
| | - | (0.007) | - | - | - | (0.036) | - | - |
| Ring Boundary II (3501–5000 ft) | - | - | 0.009 | - | - | - | −0.070 *** | - |
| | - | - | (0.007) | - | - | - | (0.027) | - |
| Ring Boundary III (5001–6500 ft) | - | - | - | −0.001 | - | - | - | −0.001 |
| | - | - | - | (0.005) | - | - | - | (0.020) |
| Bedrooms | 0.037 *** | 0.037 *** | 0.029 ** | 0.029 ** | 0.031 ** | 0.033 ** | 0.032 ** | 0.033 ** |
| | (0.012) | (0.012) | (0.012) | (0.012) | (0.015) | (0.015) | (0.015) | (0.015) |
| Bathrooms | 0.168 *** | 0.168 *** | 0.160 *** | 0.160 *** | 0.136 *** | 0.132 *** | 0.133 *** | 0.132 *** |
| | (0.042) | (0.042) | (0.040) | (0.040) | (0.020) | (0.020) | (0.020) | (0.020) |
| Age of Building | −0.007 ** | −0.007 ** | −0.006 ** | −0.006 ** | −0.003 *** | −0.004 *** | −0.003 *** | −0.004 *** |
| | (0.003) | (0.003) | (0.003) | (0.003) | (0.000) | (0.000) | (0.000) | (0.000) |
| Area | 0.000 *** | 0.000 *** | 0.000 *** | 0.000 *** | 0.000 *** | 0.000 *** | 0.000 *** | 0.000 *** |
| | (0.000) | (0.000) | (0.000) | (0.000) | (0.000) | (0.000) | (0.000) | (0.000) |
| Public Water Supply | 0.036 *** | 0.036 *** | 0.029 ** | 0.029 ** | 0.062 | 0.055 | 0.062 | 0.055 |
| | (0.014) | (0.014) | (0.012) | (0.012) | (0.048) | (0.048) | (0.048) | (0.048) |
| Distance to Biggest City (OKC or Stillwater) | 0.007 | 0.007 | 0.010 | 0.010 | 0.045 *** | 0.051 *** | 0.052 *** | 0.051 *** |
| | (0.014) | (0.014) | (0.013) | (0.013) | (0.011) | (0.010) | (0.010) | (0.010) |
| Distance to Biggest City_sq | 0.000 | 0.000 | −0.000 | −0.000 | 0.000 | 0.000 | 0.000 | 0.000 |
| | (0.000) | (0.000) | (0.000) | (0.000) | (0.001) | (0.001) | (0.001) | (0.001) |
| D_Nearest Highway | 0.000 *** | 0.000 *** | 0.000 *** | 0.000 *** | 0.000 *** | 0.000 *** | 0.000 *** | 0.000 *** |
| | (0.000) | (0.000) | (0.000) | (0.000) | (0.000) | (0.000) | (0.000) | (0.000) |
| D_Highway_sq | −0.000 *** | −0.000 *** | −0.000 *** | −0.000 *** | −0.000 *** | −0.000 *** | −0.000 *** | −0.000 *** |
| | (0.000) | (0.000) | (0.000) | (0.000) | (0.000) | (0.000) | (0.000) | (0.000) |
| D_nearest_road | 0.000 | 0.000 | 0.000 | 0.000 | −0.000 | −0.000 | −0.000 | −0.000 |
| | (0.000) | (0.000) | (0.000) | (0.000) | (0.000) | (0.000) | (0.000) | (0.000) |
| D_road_sq | −0.000 | −0.000 | −0.000 | −0.000 | −0.000 | −0.000 | −0.000 | −0.000 |
| | (0.000) | (0.000) | (0.000) | (0.000) | (0.000) | (0.000) | (0.000) | (0.000) |
| Year FE | Y | Y | Y | Y | Y | Y | Y | Y |
| Community FE | Y | Y | Y | Y | Y | Y | Y | Y |
| R-sq | 0.590 | 0.590 | 0.598 | 0.597 | 0.458 | 0.457 | 0.457 | 0.457 |
| adj. R-sq | 0.590 | 0.589 | 0.597 | 0.597 | 0.455 | 0.454 | 0.454 | 0.454 |
| N | 17,241 | 17,241 | 17,145 | 17,145 | 8227 | 8227 | 8227 | 8227 |

Robust Standard errors in parentheses: ** $p < 0.05$, *** $p < 0.01$. Note: Observations with housing prices less than USD 5000 were removed. Dark background color in the table represents the coefficients of interest (i.e., those associated with distance from housing to drilling site). Bold text represents statistically significant results.

### 5.2. Baseline Regression: The Impact of Unconventional Drilling Wells

Four variables—the nearest distance to shale well from a house and the number of unconventional drilling wells within three specific distances (0–3500, 3501–5000, and 5001–6500 feet)—are used to model the impact of unconventional drilling on housing price. Unlike the estimation results for physical characteristics and location discussed above, the estimation results for this category of variables are generally not significant. The distance between a house and its nearest shale well is statistically insignificant in both Canadian and Payne counties. Only one parameter (on the number of nearby unconventional wells between 5001 and 6500 feet) is significant, and only for Canadian County. However, considering that all other estimation results suggest no impact of unconventional wells at closer locations, it is hard to argue that increasing the number of unconventional drilling wells between 5001 and 6500 feet would lower the housing price. As we briefly noted,

environmental quality is likely not necessarily linearly related with housing price [20,22]. Further, the paper [12], examining the same issue in Pennsylvania, argued that a nonlinear relationship existed in one of the two counties in their study regions. We will discuss this issue further in the semiparametric estimation results section, following the discussion of conventional well impacts in the next section.

### 5.3. Baseline Regression: Conventional Drilling Well Impact and Additional Estimation

Unlike Pennsylvania, Oklahoma has a long history of crude oil production (Figure 4). Practically, Oklahoma residents may less pay attention to unconventional drilling wells, as some people may fail to recognize the difference between this newer type of well site and the conventional sites that they have become accustomed to seeing across the state. Thus, examining the reaction of housing prices to conventional drilling activity may be useful for discussing the possible impact of unconventional drilling. We estimate this impact using the same regression model as for unconventional drilling wells, and the results are represented in Table 4. The estimation results show that all the physical housing characteristics and locational components are very similar to those for unconventional wells in both counties. However, there is a conflicting result for the main drilling variables of interest. For Canadian County, there is a significant negative effect associated with the nearest well distance variable. Although other measures of drilling activity (counts of wells within the three ring boundaries) do not show a significant effect, this is in direct contrast to the results from the unconventional wells in Table 3, where no impacts were found. It is also the opposite result shown for conventional wells in Payne County, where housing prices increase as distance to the nearest well grows. Thus, it appears that home buyers in Canadian County will pay a premium to be closer to a conventional well, while home buyers in Payne pay a premium to be further away. We will discuss this result further in the semiparametric estimation results section that follows.

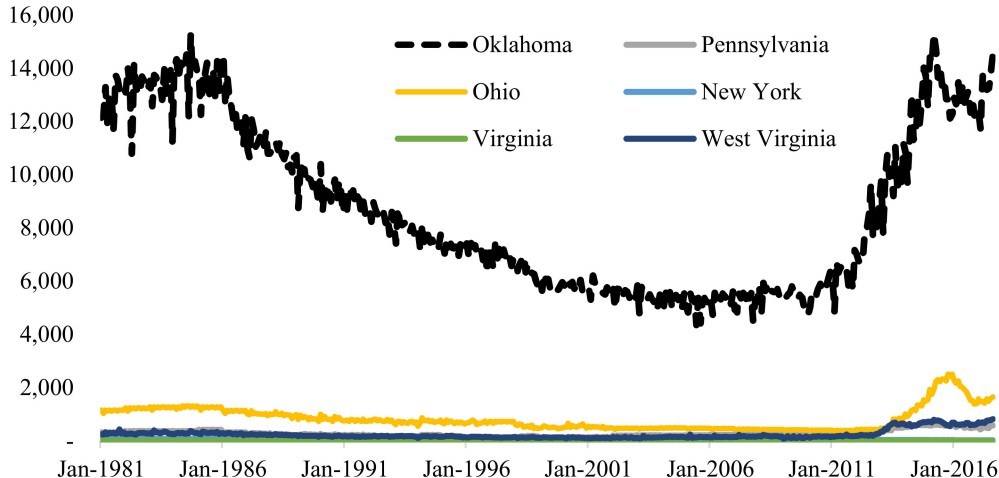

**Figure 4.** Crude oil production trend in OK, OH, PA, NY, VA, and WV. Note: The unit of vertical axis is a thousand barrels. Source: U.S. Energy Information Administration (https://www.eia.gov/dnav/pet/pet_crd_crpdn_adc_mbbl_m.htm, accessed on 10 December 2019).

We note, however, that the distance between a house and the nearest unconventional well is over 40 miles in some instances. Intuitively, being located over 10 miles from a well should not have any impact on housing price, since the well equipment visibility is negligible at that distance and potential environmental impacts are also minimal. Taking this into account, we estimate several additional models that apply this intuition. To achieve this, we forcefully allocate the distance variable as missing if the nearest distance between a house and unconventional well is larger than 10 miles. All of the physical housing parameters are comparable to those in the original models. In addition, the nearest distance to unconventional drilling wells is not significant in any of the cases.

### 5.4. Semiparametric Estimation Results

The estimation results for the linearly modelled portion of the semiparametric approach are reported in Table 5. As expected, all results for the housing characteristics are very close to those for the baseline regression results in Tables 3 and 4. Now, we switch to a further discussion of the drilling impact, assuming some unknown function represents the relationship between housing price and distance to the nearest well. Previous studies argue for the possibility of a nonlinear relationship between an environmental pollutant source and a housing price [20,22]. The study [12] takes this a step further and uses semiparametric estimation to provide evidence of a nonlinear relationship between shale drilling activity and property value in their study region in Pennsylvania. Figures 5 and 6 show the semiparametric results for the effect of distance to the nearest conventional (or unconventional) well on the log of housing price. For Canadian County (Figure 5), the distributions of both unconventional drilling and conventional drilling wells make it difficult to distinguish whether a nonlinear relationship exists with the log of housing price. Further, the flatness of the two curves generally suggests that housing price does not appear to be impacted by the distance to the nearest well—either conventional or unconventional. For Payne County (Figure 6), the results show an even more vague relationship. Thus, the semiparametric results for both counties do not show a clear relationship between housing price and distance to drilling site. We provide the Hardle–Mammen test results in Table 5. The null hypothesis for this test is that the parametric and nonparametric fits are not different, and rejecting the null implies that a polynomial adjustment is suitable, rather than a linear specification. Except for one case (conventional drilling well in Canadian County), we reject the null hypothesis at the $p = 90\%$ level; thus, we may argue for a nonlinear relationship between a housing price and the nearest drilling site in most cases. Moreover, the test statistics from both types of wells in Payne County are strongly rejected at the 99% level. Based on this, we argue that we do not find consistent and significant evidence to suggest that drilling activity significantly influences nearby housing prices in both Canadian and Payne counties.

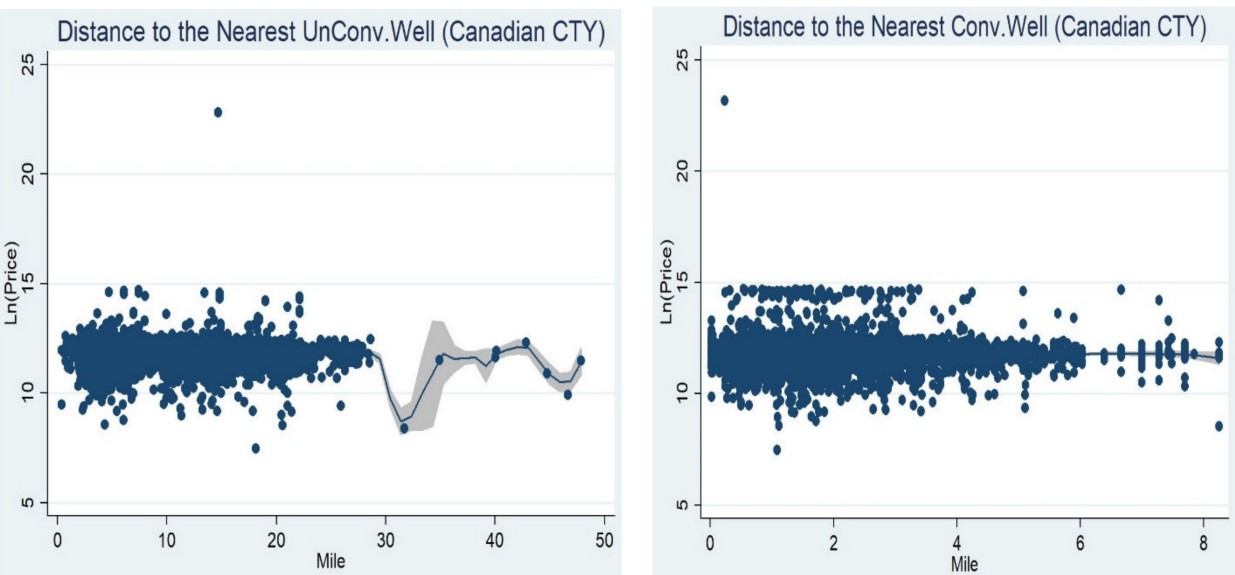

**Figure 5.** Semiparametric results for Canadian County. Note: 1. Both figures represent the relationship between the logarithm of housing price and the distance between a house and its nearest well (unconventional drilling well, left; conventional drilling well, right). 2. Horizontal axes are distance between a house and its nearest well. Vertical axes are logarithmic house sale prices. 3. The shaded areas indicate the computed 95% confidence interval.

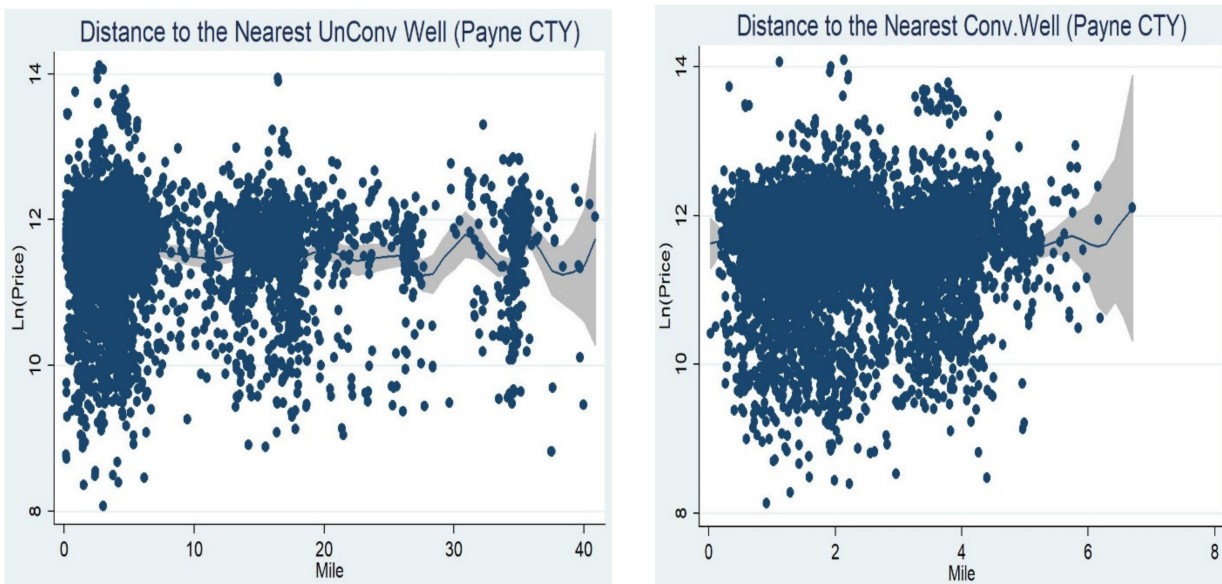

**Figure 6.** Semiparametric results for Payne County. Note: 1. Both figures represent the relationship between the logarithm of housing price and the distance between a house and its nearest well (unconventional drilling well, left; conventional drilling well, right). 2. Horizontal axes are distance between a house and its nearest well. Vertical axes are logarithmic house sale prices. 3. The shaded areas indicate the computed 95% confidence interval.

**Table 5.** Results of linear part in semiparametric estimation.

| Ln (Sale Price) | Canadian County | | Payne County | |
|---|---|---|---|---|
| | **Unconventional Drilling Well** | **Conventional Drilling Well** | **Unconventional Drilling Well** | **Conventional Drilling Well** |
| Bedrooms | **0.034 ***** | **0.035 ***** | **0.034 **** | **0.032 **** |
| | (0.012) | (0.012) | (0.016) | (0.015) |
| Bathrooms | **0.160 ***** | **0.166 ***** | **0.161 ***** | **0.136 ***** |
| | (0.037) | (0.040) | (0.020) | (0.020) |
| Age of Building | **−0.006 **** | **−0.006 **** | **−0.003 ***** | **−0.003 ***** |
| | (0.003) | (0.003) | (0.000) | (0.000) |
| Area | **0.000 ***** | **0.000 ***** | **0.000 ***** | **0.000 ***** |
| | (0.000) | (0.000) | (0.000) | (0.000) |
| Public Water Supply | **0.037 ***** | **0.033 **** | 0.033 | 0.058 |
| | (0.014) | (0.013) | (0.054) | (0.048) |
| Distance to Biggest City (OKC or Stillwater) | 0.018 | 0.006 | **0.055 ***** | **0.055 ***** |
| | (0.013) | (0.014) | (0.012) | (0.011) |
| Distance to Biggest City_sq | −0.000 | 0.000 | −0.000 | −0.000 |
| | (0.000) | (0.000) | (0.001) | (0.001) |
| D_Nearest Highway | **0.000 ***** | **0.000 ***** | **0.000 ***** | **0.000 ***** |
| | (0.000) | (0.000) | (0.000) | (0.000) |
| D_Highway_sq | **−0.000 ***** | **−0.000 ***** | **−0.000 ***** | **−0.000 ***** |
| | (0.000) | (0.000) | (0.000) | (0.000) |
| D_nearest_road | 0.000 | 0.000 | −0.000 | 0.000 |
| | (0.000) | (0.000) | (0.000) | (0.000) |
| D_road_sq | −0.000 | −0.000 | −0.000 | −0.000 |
| | (0.000) | (0.000) | (0.000) | (0.000) |
| Year FE | Y | Y | Y | Y |
| Community FE | Y | Y | Y | Y |
| R-sq | 0.563 | 0.596 | 0.381 | 0.418 |
| adj. R-sq | 0.562 | 0.595 | 0.377 | 0.415 |
| N | 16,846 | 16,852 | 6728 | 8225 |
| Critical value (95%): 1.96 | | | | |
| Critical value (90%): 1.645 | | | | |
| Hardle–Mammen Test Statistics | **1.824** | 0.787 | **3.792** | **8.900** |

Robust Standard errors in parentheses: ** $p < 0.05$, *** $p < 0.01$. Bold text represents statistically significant results.

## 6. Conclusions

In this study, we examined the existence of externalities from unconventional drilling on housing prices in two central Oklahoma counties. Additionally, taking into considering the long tradition of resource mining in Oklahoma, we examined the impact from conventional drilling activity. Housing values are also affected by their locational environment, such as the size of the city [26]. Our two study locations may show the differences between 'Built-Up' areas (Canadian County) and small college towns (Payne County). The empirical results are consistent with prior hedonic models in terms of the influence of physical housing characteristics and proximity to cities and highways. However, the results for drilling activity find minimal significant effects across a variety of specifications. To examine a possible nonlinear relationship between housing price and drilling site, we provide a semiparametric estimation. The results from this empirical strategy support the lack of consistent evidence demonstrating a measurable effect of nearby unconventional drilling activity on housing price. These results are in conflict with those from several earlier studies [11,14,15]. However, our results are in line with [12], which is the only study to use a nonlinear empirical specification on this topic to date. One possibility for the lack of significance associated with any of the unconventional drilling variables is that countervailing impacts are at work: negative ones associated with environmental or scenic concerns and positive ones associated with possible gains from mineral rights. This study did not explicitly consider such mineral rights, which have been shown to be substantial [27]. It may be that potential gain from mineral rights may have a positive influence on housing price.

Several limitations exist for this study. Although we compare the results between unconventional drilling and conventional drilling, these are aspects associated with shale well activity that are not fully accounted for in our model. The paper [14] argues that a prospective consumer may become aware of shale drilling through one of four paths—online open database, increased truck traffic, noise from drilling, or the visible aspect of drilling. Their study region is Pennsylvania, which has experienced a rapid increase in unconventional drilling since the mid-2000s; however, this state does not have a comparatively long history of conventional crude oil mining. Because of this lack of history with conventional drilling, a prospective home buyer in Pennsylvania may be wary of environmental changes associated with shale gas drilling. Alternatively, Oklahoma has a long tradition of mining crude oil, and local residents may be accustomed to the sights and sounds of drilling activity or may be unable to distinguish between conventional and unconventional sites. An alternative way of making this point is to assert that people in Oklahoma have less sensitivity to unconventional drilling activity when compared to Pennsylvania residents. For example, paper [13] studied people's risk-averting behavior associated with unconventional shale drilling. Using consumer's water bottle purchasing data in Pennsylvania, they found that expenditures associated with risk-averting practices in Pennsylvania shale region were larger than 19 million USD for the year 2010. This implies that the extent of the impact may be comparatively small in Oklahoma than Pennsylvania, and this may be one reason for the lack of results in this analysis.

The empirical results from this study suggest that the effect of unconventional drilling does not have a clear impact on housing prices in central Oklahoma. However, the boom of unconventional drilling started less than 10 years ago (particularly for the counties considered here). The (lack of) results call for additional study over a longer time period, and across more areas, to observe potentially changing effects from unconventional drilling. In terms of policy implications, the minimal impact of drilling on housing prices in these two Oklahoma counties suggests that revamping property tax policy out of concern for oil and gas effects is likely not necessary. Our findings also agree with [12], who highlight the importance of study area and stress that a variety of econometric techniques should be used to test robustness.

**Author Contributions:** Conceptualization, K.L. and B.W.; methodology, K.L. and B.W.; software, K.L.; formal analysis, K.L.; investigation, K.L.; writing—original draft preparation, K.L.; writing—review and editing, K.L. and B.W.; visualization, K.L.; supervision, B.W. All authors have read and agreed to the published version of the manuscript.

**Funding:** This research received no external funding. The APC was funded by Changwon National University.

**Data Availability Statement:** The data presented in this study are available on request from the corresponding author. The data are not publicly available due to privacy issues (County Assessor offices typically sell their housing purchase data for a fee).

**Acknowledgments:** This study is a revised version of the second chapter of Kangil Lee's Ph.D. dissertation.

**Conflicts of Interest:** The authors declare no conflict of interest.

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
