# Peer review of "A Study on the Impact of Unconventional (and Conventional) Drilling on Housing Prices in Central Oklahoma"

_sustainability, doi:10.3390/su132413880_

Round 1

Reviewer 1 Report

The paper presents a robust analysis, supported by data and results compatible with the objectives. The methodology used is interesting and lends itself to further study in the future. The paper is sufficiently complete in all its parts. However, I suggest expanding the literature review. I recommend reading the following article:

  • F., Tavano. D., De Ruggiero. M.: Hedonic price of the built-up area appraisal in the Market Comparison Approach, Smart Innovation, System and Technologies (2190-3018), SPRINGER, (2021).

It deals with the influence of location on real estate values.

Author Response

Response to Reviewer 1

[General Comment]

The paper presents a robust analysis, supported by data and results compatible with the objectives. The methodology used is interesting and lends itself to further study in the future. The paper is sufficiently complete in all its parts.

Response: Thank you for your generous comment.

[Minor Comment]

I suggest expanding the literature review. I recommend reading the following article: F., Tavano. D., De Ruggiero. M.: Hedonic price of the built-up area appraisal in the Market Comparison Approach, Smart Innovation, System and Technologies (2190-3018), SPRINGER, (2021).

Response: We add the paper you suggested in footnote 10 on page 15 (Please find the highlighted sentence; the number of the paper is 26 in Reference)

Reviewer 2 Report

The paper provides an original and interesting topic and the title express clearly the object of the research. The overall structure is well constructed and has no weak points by scientific and conceptual point of views. Only an expansion of the literature is suggested ; among other authors :  Del Giudice, V., De Paola, P., Manganelli, B., Forte, F. ‘The Monetary Valuation of Environmental Externalities through the Analysis of Real Estate Prices’. Sustainability 2017, 9, 229. https://doi.org/10.3390/su9020229

Author Response

Response to Reviewer 2

[General Comment]

The paper provides an original and interesting topic and the title express clearly the object of the research. The overall structure is well constructed and has no weak points by scientific and conceptual point of views.

Response: Thank you for your favorable comment.

[Minor Comment]

Only an expansion of the literature is suggested; among other authors :  Del Giudice, V., De Paola, P., Manganelli, B., Forte, F. ‘The Monetary Valuation of Environmental Externalities through the Analysis of Real Estate Prices’. Sustainability 2017, 9, 229. https://doi.org/10.3390/su9020229

Response: We add the paper you suggested on page 1 (Please find the highlighted sentence; the number of the paper is 10 in Reference)

Reviewer 3 Report

The article that was revised is very attractive both from scientific and popularizing point of view. I found it well structured, with arguments logically presented. I evaluate the paper very positively. Authors were 'scientifically distrustful' and because of that - looking for the answers deeply and persistantly. Authors showed very good level of scientific worshop and chose the methods adequatly to the problem. 

I have only one hint - not as a necessity but just for taking into consideration. It is about the title of the article. Right now it is proposed as: 'The Impact of Unconventional Drilling on Housing Prices in Central Oklahoma'

Please, note that such a title suggests the readers (subconsciously) that the impact between unconventional drilling on housing prices in Central Oklahoma was really observed (and was proven by Authors). The title sounds like a kind of scientific promise which is confusing somehow. To be fair and more adequate to the conclusions formulated in the paper - in my opinion the Authors should modify the title. My humble proposal is following:

'The Study on Impact of Unconventional Drilling on Housing Prices in Central Oklahoma'

Author Response

Response to Reviewer 3

[General Comment]

I found it well structured, with arguments logically presented. I evaluate the paper very positively. … Authors showed very good level of scientific workshop and chose the methods adequately to the problem.

Response: Thank you for your favorable comment.

[Minor Comment]

I have only one hint - not as a necessity but just for taking into consideration. It is about the title of the article. Right now it is proposed as: 'The Impact of Unconventional Drilling on Housing Prices in Central Oklahoma'. Please, note that such a title suggests the readers (subconsciously) that the impact between unconventional drilling on housing prices in Central Oklahoma was really observed (and was proven by Authors). The title sounds like a kind of scientific promise which is confusing somehow. To be fair and more adequate to the conclusions formulated in the paper - in my opinion the Authors should modify the title. My humble proposal is following:

'The Study on Impact of Unconventional Drilling on Housing Prices in Central Oklahoma'

Response: Taking your comment, we changed the title of the paper. (Please find the highlighted title on page 1)
